# Effects of Far-Infrared Rays Emitted from Loess Bio-Balls on Lymphatic Circulation and Reduction of Inflammatory Fluids

**DOI:** 10.3390/biomedicines12102392

**Published:** 2024-10-19

**Authors:** Yong Il Shin, Min Seok Kim, Yeong Ae Yang, Gye Rok Jeon, Jae Ho Kim, Yeon Jin Choi, Woo Cheol Choi, Jae Hyung Kim

**Affiliations:** 1Department of Rehabilitation Medicine, School of Medicine, Pusan National University, Yangsan 50612, Republic of Korea; rmshin@pusan.ac.kr; 2Monash Health, Melbourne, VIC 3800, Australia; minseok.kim@monashhealth.org; 3Department of Occupational Therapy, Room 411, Seongsan Hall (Bldg. F), Inje University, 197 Inje-ro, Gimhae-si 50834, Republic of Korea; otyya62@inje.ac.kr; 4R&D Center, eXsolit, Yangsan-si 50611, Republic of Korea; grjeon@pusan.ac.kr (G.R.J.); jhkim@pusan.ac.kr (J.H.K.); 5R&D Center, Hanwool Bio, Seokgyesandan 6-gil, Yangsan-si 50516, Republic of Korea; bio1004@naver.com (Y.J.C.); lih1769@naver.com (W.C.C.)

**Keywords:** loess bio-ball, far infrared rays (FIR), lymphatic circulation, inflammatory fluids, bio-impedance

## Abstract

**Background**: FIR therapy is used in various medical settings to treat diseases associated with inflammation and edema. Unlike conventional FIR lamp therapy, this study investigated how body fluids change depending on the intensity and duration of FIR irradiation to the whole body. **Method**: Subjects in group A (*n* = 27) were exposed to FIR emitted from a loess bio-ball mat set at 40 °C for 30 min, and subjects in group B (*n* = 27) were exposed to FIR emitted from a loess bio-ball mat set at 30 °C for 7 h during sleep. Changes in bioimpedance parameters and fluid-related values were measured using a body fluid analyzer before and after exposure to FIR. **Results**: Changes in bioimpedance parameters associated with inflammatory fluids were quantitatively confirmed. In group A, there was a minimal change in fluid-related measurements. However, significant changes in bioimpedance parameters associated with inflammatory fluids were observed in group B exposure to FIR for 7 h during sleep. **Conclusions**: FIR emitted from loess bio-balls activates biological tissues and lymphatic circulation, gradually reducing the levels of inflammatory fluids over time.

## 1. Introduction

The peak wavelength of FIR emitted from loess bio-balls is 9.5~9.8 μm, which closely matches the vibration wavelength of water molecules in the human body [1]. The FIR emitted from loess bio-balls penetrates the epidermis and adipose tissue and transmits radiation energy in the form of vibrational energy to the water molecules in the human body. This can increase the body’s core temperature and activate water molecules in the cell membranes and tissues [2]. The FIR emitted from loess bio-balls can be easily applied in various forms for therapeutic purposes in a medical setting. The FIR emitted from loess bio-balls is expected to not only increase the body’s core temperature, but also activate cells and tissues, and further stimulate lymphatic circulation to remove waste products and lower inflammation levels [1].

Blood circulates from the arteries through the capillaries to the veins, leaving behind some blood components called interstitial fluid (ISF), or fluid between cells. Interstitial fluid containing metabolic waste products, toxins, viruses, and bacteria is collected in lymph capillaries. This lymph fluid then flows through lymphatic vessels, waste products are filtered through the lymph nodes, and viruses and bacteria are processed by the immune cells that reside in the lymph nodes. The lymphatic system collects excess fluid, proteins, and toxins from cells and tissues, and then returns them to the bloodstream. However, if lymphatic circulation is compromised, inflammatory fluid accumulates in the body, causing swelling and edema [3]. Lymphatic vessels drain lymph by utilizing external forces exerted on the vessel walls by the surrounding tissues (extrinsic mechanisms) and gentle contractions of lymphatic muscle cells that collect lymph (intrinsic mechanisms) [4].

Bioimpedance analysis is a rapid, noninvasive method for estimating body composition and assessing tissue resistance and cell membrane capacitance [5]. Bioelectrical impedance (*Z*) is a vector composed of two independent components: real resistance (*R*) and imaginary capacitive reactance (*Xc*), as shown in Figure 1. These two values can be used to calculate the phase angle (PA), which is used to evaluate the quality of the cell membrane and the nutritional status of the cell [6]. PA represents the angle formed by Z between *R* and *Xc* at a given frequency, and is calculated as the arctangent of the radius between the resistance and reactance ($PA={[tan}^{-1}$($X_{C}/R$)] × 180°/π) [7,8].

Bioimpedance analysis measures the resistance of body tissues to electrical currents and provides insight into a variety of physiological conditions. Resistance is coupled to resistive pathways through fluids within tissues, and reactance is coupled to capacitive pathways such as structures in cell membranes [9]. At 5 kHz, a low-energy current cannot pass through the cell membrane and thus flows into the extracellular water (ECW; indicated as *R_e_*) in the equivalent circuit, as shown in Figure 2 [10]. Here, I is the current through both the ECW and ICW, I_1_ is the current passing through only the ECW, I_2_ is the current passing through both the cell membrane and ICW, *R_e_* is the resistance of the ECW, *R_i_* is the resistance of the ICW, and *R_m_* and *C_m_* are the resistance and capacitance of the cell membrane, respectively. From 50 kHz, the current is strong enough to penetrate the cell membrane and flow into both the ECW and intracellular water (ICW; indicated as *R_i_*). The bioimpedance values for the current passing through the ECW at low frequencies and for the current flowing through the cell membrane and ICW at frequencies above 50 kHz can be obtained using a multi-frequency impedance analyzer [11]. By applying these bioimpedance values and individual physical characteristics to a mathematical formula [12], various information, such as body composition and body water, can be obtained [13,14]. Total body water (TBW) accounts for approximately 60% of body weight, depending on the subject’s age, sex, and body mass index (BMI). ICW accounts for approximately 40% of TBW and ECW accounts for approximately 20% of TBW. ISF accounts for approximately 15% of ECW, whereas plasma accounts for 5% of ECW. ISF has a composition similar to that of plasma despite its low protein content. The cells that make up human tissue are composed of ECW and ICW, which act as electrical conductors, and the cell membrane acts as an electrical capacitor [9,10].

The BIA is a valuable tool for assessing the health status of cells and tissues. It provides a variety of useful biological parameters using mathematical formulas [15]. BIA has been used to diagnose various diseases [16]. It has also been used to evaluate hydration status, body composition, muscle-to-fat ratio, degree of obesity, muscle mass balance, edema, and nutritional status [17,18]. Multi-frequency BIA can be used to detect changes in extracellular fluid volume during the diagnostic evaluation of lymphedema [19]. Changes in bioimpedance measurements reflect the levels of inflammatory fluids in the body in various diseases [20,21,22]. Bioimpedance is used in the perioperative setting to quantitatively assess hydration status and fluid distribution in patients undergoing acute high-risk abdominal (AHA) surgery [20]. As changes in bioimpedance parameters are associated with fluid balance, weight changes, and postoperative complications, fluid status assessed by BIA provides useful information for fluid management in patients undergoing AHA surgery. The relationship between changes in bioimpedance parameters and response to fluid therapy has been identified in critically ill patients during the early postoperative period [21]. Overhydration can negatively impact post-operative morbidity and mortality in these patients, and the cut-off values for overhydration status can be determined using the ECW ratio (ECW/TBW). PA, calculated from bioelectrical parameters, is a potential tool for identifying the inflammatory state of individuals with cardiovascular disease (CVD) [22]. PA is inversely proportional to inflammatory markers, such as C-reactive protein (CRP) and tumor necrosis factor-α (TNF).

In this study, the FIR emitted from functional loess bio-balls was applied to experimental subjects in the early stages of inflammation and edema (stage 0 or 1). Subjects in group A were exposed to FIR emitted from a loess bio-ball mat set at 40 °C for 30 min, and subjects in group B were exposed to FIR emitted from a loess bio-ball mat set at 30 °C for 7 h during sleep. Changes in bioimpedance parameters associated with inflammatory fluids were measured using a body fluid analyzer. The bioimpedance parameters associated with inflammation and lymphedema in the early stages were investigated under different conditions (FIR intensity and duration).

## 2. Materials and Methods

### 2.1. Selective Absorption Between FIR Emitted from Loess Bio-Ball and Water Molecules

Unlike loess and loess balls that are heat-treated at high temperatures of over 1000 °C, loess bio-balls manufactured using a low-temperature drying method retain the unique properties of the original loess [1]. Bio-balls emit a large amount of FIR at wavelengths of 5–20 μm at 40 °C (313 K). The radiant intensity was $3.74\times 10^{2\;}W/m^{2}$ around 9.5–9.8 μm. These FIRs are emitted by Si-O stretching motions (Si-O deformation, Si-O stretching, and Si-O bending) in loess bio-balls. The wavelength (5–20 μm) of the radiant intensity emitted from the loess bio-balls matches the absorption wavelength band known as the growth line (5.6–14 μm), which is essential for the survival of natural organisms [2,23]. Therefore, FIR is selectively absorbed by water molecules in the body by wave resonance and then converted into thermal energy, while the rest is transferred as vibrational energy to activate the surrounding biological tissues [24,25].

### 2.2. Study Participants

Subjects for the clinical study on inflammation and lymphatic circulation were recruited through an article in the local newspaper “Yangsan News Park” dated 7 November 2023. The study participants were over 30 years of age and had either been diagnosed with inflammation or edema at a medical institution or experienced discomfort due to lymphatic circulation disorders. These participants complained of discomfort due to mild inflammation or swelling in the early stages of lymphedema stage 0 (the affected area was swollen, tight, and heavy, but there were no external signs of swelling) or stage 1 (occasional swelling that disappeared when the affected area was lifted) [3]. Of the 70 participants, 54 were finally divided into group A (30 min at 40 °C) and group B (7 h of sleep at 30 °C) after considering their health status and willingness to participate. None of the participants were taking medications that could affect autonomic nervous system (ANS) function. The demographic characteristics and physical condition of the participants in each group are shown in Table 1.

### 2.3. Trial Design and Setting

This experimental protocol was approved by the Inje University Bioethics Committee (registration number: INJE 2023-05-035-005) dated 20 September 2023, under the clinical trial registration titled “Improvement of blood circulation and health promotion effects in related diseases by FIR emitted from loess bio-balls”. This experiment was mainly conducted based on the Consolidated Standard of Reporting Trials (CONSORT), as shown in Figure 3, at Hanwool Bio’s Research and Development Center, and included self-reported home measurements from study participants. The exclusion or discontinuation criteria for the intervention were as follows: pregnant women or applicants with cancer, diabetes, cardiovascular, or neurological diseases. Participants who showed symptoms of mental weakness or anxiety, or who withdrew from participation due to illness or personal reasons were also excluded from the intervention. Of the 70 participants, 6 were excluded because they did not meet the inclusion criteria (*n* = 4) or withdrew from the study for personal reasons (*n* = 2). After hearing an explanation of each experiment, 64 participants expressed their willingness to participate in each experiment, taking into account their personal circumstances. The participants freely chose experimental group A or experimental group B, and 32 subjects were assigned to each group. During the follow-up experiment, 10 additional subjects were excluded due to illness or personal reasons: five from group A and five from group B. The total number of experimental subjects who participated in the experiment was 54, and they were divided into group A (*n* = 27) and group B (*n* = 27) and participated in each experiment.

A loess bio-ball mat and Jangsoo bio-ball bed (7111, Jangsoo Industry Co., Ltd., Seoul, Republic of Korea) were used as energy sources for the FIR. Loess bio-balls emit a large amount of FIR at 5–20 μm (peak wavelength: 9.5–9.8 μm) and the radiant energy was $3.74\times 10^{2}\;W/m^{2}$. Bioimpedance measurements to detect changes in PA and inflammatory fluid-related values due to FIR emitted from loess bio-balls were performed using a body water analyzer (InBody S10, InBody, Seoul, Republic of Korea) as follows: the study participants lay comfortably on a loess bio-ball mat with clamp-type electrodes attached to both ankles and middle fingers. In group A, 27 participants were exposed to FIR (approximately $3.72\times 10^{2}$ W/$m^{2}\cdot \mu m$) emitted from a loess bio-ball mat set at 40 °C, and bioimpedance parameters and fluid-related changes were measured before exposure and after 30 min of exposure. In group B, 27 participants were exposed to FIR (approximately $3.55\times 10^{2}$ W/$m^{2}\cdot \mu m$) emitted from a loess bio-ball mat set at 30 °C during sleep, and changes in bioimpedance parameters and body water-related values were measured before exposure and after 7 h of exposure. To investigate the changes in inflammatory fluid caused by FIR during sleep, the set temperature applied to the mat was lowered to 30 °C to ensure that the participants in group B could sleep comfortably.

### 2.4. Statistical Analyses

All statistical analyses for groups A (*n* = 27) and B (*n* = 27) were performed using IBM-SPSS Statistics (version 29.0.2.0; IBM Corp., SPSS Inc., Armonk, NY, USA). Pearson correlation coefficients (r) and *p*-values were calculated for the measured bioimpedance parameters and fluid-related variables in each group before and after FIR application. A *p*-value of <0.001 was considered significant. Data processing, graphing, and logistic fitting were performed using Microsoft Excel 2016 (Microsoft Corp., Redmond, WA, USA).

## 3. Results

### 3.1. Changes in Phase Angle and Fluid-Related Values Before and After Irradiation of FIR Emitted from Loess Bio-Balls

Because the physiology and pathology of tissues change with tissue health, bioimpedance values also change depending on whether the tissue is healthy or diseased. Owing to its diverse biological effects, FIR provides therapeutic benefits, such as anti-inflammatory, immunomodulatory, and antioxidant stress effects [26]. FIR can effectively improve lymphedema, reduce limb circumference, and reduce the thickness of skin and subcutaneous lymphedema tissue, thereby reducing the recurrence rate of lymphedema and improving the patient’s quality of life [27]. Bioimpedance reflects tissue health including inflammation, swelling, and infection, and these changes can be analyzed using various bioimpedance parameters [17].

Table 2 shows the changes in the phase angle (PA) and inflammatory fluid-related values before and after applying the FIR emitted from the loess bio-ball mat. The measurement values in the second and third rows from the top indicate the changes in PA, body fluids (ICW and ECW), and inflammatory fluid-related values before and after 30 min of exposure to FIR emitted from the loess bio-ball mat set at 40 °C. There was minimal or no change in these values before and after applying FIR. In the fourth row, Delta FIR (40 °C for 30 min) shows the difference (including percentages in parentheses) between these values before and after applying FIR. This confirms that there were virtually no fluid-related changes in the body of 27 experimental subjects when exposed to FIR under these conditions (40 °C for 30 min). The fifth and sixth rows indicate the changes in PA, body fluids (ICW and ECW), and body fluid-related values before and after applying FIR emitted from a loess bio-ball mat set at 30 °C for 7 h during sleep. In the seventh row, Delta FIR (30 °C for 7 h) shows the difference (including percentages in parentheses) between these values before and after applying FIR. There was a significant increase in PA (+3.80%) and a marked decrease in inflammatory fluid-related measurements, such as ECF (−3.38%), ECW/ICW (−4.84%), ECW/TBW (−2.63%), and ECW/BCM (−2.33%). This suggests that the decrease in body fluid (especially ECW) in slow-moving lymphatic circulation is closely related to the duration of FIR exposure. The Pearson correlation coefficients before FIR and after 30 min of exposure to FIR at 40 °C were: 0.963 (PA), 0.997 (ECW), 0.844 (ECW/ICW), 0.730 (ECW/TBW), and 0.858 (ECW/BCM). The Pearson correlation coefficients before FIR and after 7 h of exposure to FIR at 30 °C were: 0.989 (PA), 0.997 (ECW), 0.816 (ECW/ICW), 0.847 (ECW/TBW), and 0.846 (ECW/BCM). All *p*-values for the bioimpedance parameters measured before and after FIR application were less than 0.001.

### 3.2. Change in PA Before and After FIR Exposure from Loess Bio-Balls

PA measures cellular integrity and health and is related to body cell mass and hydration. Therefore, it may indirectly reflect inflammation [28]. PA represents the ratio of *Xc* to *R*, and during inflammation, PA decreases because *Xc* decreases more than *R* [29]. PA is also considered an indicator of cellular health, with higher values reflecting better cellularity, cell membrane integrity, and cell function [30]. In clinical practice, PA is an independent prognostic indicator for patients with breast cancer and is used as an indicator of health status and functional ability in breast cancer survivors [31]. The cutoff value of 5.6° may be an indicator of health status and functional ability in breast cancer survivors [32].

Table 2 shows the changes in the PA before and after applying the FIR emitted from the loess bio-balls. PA slightly increased from 5.83 ± 0.63 [°] to 5.86 ± 0.74 [°] by 0.51% before and after applying FIR emitted from the loess bio-ball mat set at 40 °C for 30 min. However, there was a significant difference in PA changes before and after exposure to FIR for 7 h on a mat set at 30 °C during sleep. This value increased by 3.8% from 6.06 ± 0.79 [°] before the FIR to 6.29 ± 0.76 [°] after the FIR. The increase in PA values reflects the improvement in cell membrane function and the reduction in inflammatory fluids due to FIR released from the loess bio-balls during sleep [29].

### 3.3. Alteration in ECW Before and After FIR Irradiation from Loess Bio-Balls

ECW contains waste products and inflammatory components derived from cellular metabolism. Edema is an abnormal accumulation of ECW resulting from the dysfunction of physiological mechanisms that maintain proper concentrations of body fluid, circulating intravascular volume, and cellular and extracellular electrolytes [33]. Dysfunction of these systems can lead to edema through one of two final pathways: (1) excessive fluid filtration through capillaries or (2) inadequate drainage of ISF by lymphatic vessels.

Table 2 shows the changes in ECW before and after the application of FIR emitted from the loess bio-ball mat. Before and after 30 min of exposure to FIR emitted from a mat set at 40 °C, ECW slightly increased (+0.08%) from 12.69 ± 2.34 [L] to 12.70 ± 2.40 [L]. However, ECW before and after 7 h of exposure to FIR on a mat set at 30 °C during sleep significantly decreased by 3.38%, from 14.22 ± 3.07 [L] to 13.74 ± 3.02 [L]. Unlike blood circulation, which quickly flows throughout the body in approximately 1 min, lymph circulates at a very slow speed of 280 to 1350 μm/s (average speed: 0.9 mm/s) within lymphatic vessels [34,35]. Therefore, it is estimated that ECW containing waste products released from the cellular energy metabolism were further reduced when exposed to FIR for 7 h on a mat set at 30 °C. Wentian et al. reported that FIR treatment reduced ECF in 32 patients with chronic lymphedema, and that this treatment could effectively reduce water, lipids, proteins, and hyaluronic acid accumulated in swollen tissues, thereby improving edema symptoms in the limbs [36].

### 3.4. Relationship Between PA and ECW/ICW Before and After FIR Irradiation from Loess Bio-Balls

Edema is accompanied by fluid redistribution between the ECW and ICW spaces and may impair cell function. PA is also one of the best indicators of cell membrane function and is associated with the ECW/ICW ratio [37]. A low PA corresponds to a high ECW/ICW ratio in systemic diseases with ECW expansion and ICW loss [38]. One of the most important confounding factors is swelling of the distal extremities, which can lead to lymphedema [39].

Figure 4 shows the relationship between PA and ECW/ICW before and after applying FIR to the 27 research subjects. The blue circles represent the values measured before the subjects were exposed to FIR, and the orange circles represent the values measured after the subjects were exposed to FIR. The blue and orange dotted lines represent the estimated lines of the measurements for the 27 subjects. Figure 4a shows the relationship between PA and ECW/ICW before and after applying FIR for 30 min on a loess bio-ball mat set at 40 °C. The mean values (0.61, 5.83) before exposure to FIR were similar to those observed 30 min after exposure to FIR (0.61, 5.86). Figure 4b shows the relationship between PA and ECW/ICW before and after exposure to FIR for 7 h on a loess bio-ball mat set at 30 °C during sleep. There was a significant difference between the mean values (0.62, 6.06) before exposure to FIR and those (0.59, 6.29) after 7 h of exposure to FIR during sleep, which resulted in a 4.84% decrease in ECW/ICW and a 3.80% increase in PA. The *p*-values for PA and ECW/ICW before and after FIR application were less than 0.001. In Figure 4b, the measured values (blue and orange circles) and the estimated dotted lines (blue and orange) shift significantly to the left and slightly upward after applying FIR for 7 h. This suggests that long-term exposure to FIR emitted from loess bio-balls during sleep can improve cell membrane function and reduce ECW, an inflammatory fluid containing waste products released from cells, thereby alleviating edema.

### 3.5. Relationship Between PA and ECW/TBW Before and After FIR Irradiation from Loess Bio-Balls

ECW/TBW is clinically recognized as an indicator of edema. Edema formation occurs due to fluid redistribution between the ECW and ICW spaces and can damage cellular function [40]. ECW/TBW also predicted the survival of patients with cancer and sarcopenia. It has been reported that in cancer patients with sarcopenia, mortality increases rapidly when ECW/TBW exceeds a threshold (ECW/TBW ≥ 0.385) [41].

Figure 5 shows the relationship between PA and ECW/TBW before and after applying the FIR emitted from loess bio-balls to 27 subjects. Figure 5a shows the relationship between PA and ECF/TBW before and after applying FIR for 30 min from mat set at 40 °C. The mean values before and after FIR exposure showed little change, at 0.38. Figure 5b shows the relationship between PA and ECW/TBW before and after 7 h of exposure to FIR on a loess bio-ball mat set at 30 °C. The mean value of ECW/TBW decreased by 2.63% from 0.38 ± 0.01 before FIR to 0.37 ± 0.01 after FIR. The *p*-values for PA and ECW/TBW before and after FIR application were less than 0.001. This suggests that exposure to FIR emitted from the loess bio-ball mat during sleep can improve cell membrane function and reduce inflammatory fluid levels, thereby alleviating early-stage inflammation and edema.

### 3.6. Relationship Between PA and ECW/BCM Before and After FIR Irradiation from Loess Bio-Balls

ECW/BCM is a sensitive biomarker of fluid overload and malnutrition. Increased ECW/BCM ratio is independently associated with an increased risk of functional impairment in patients undergoing maintenance hemodialysis. Patients with increased ECW/BCM are more likely to develop complications, such as pulmonary edema, heart failure, and lower extremity edema, all of which are associated with reduced physical activity and function [42].

Figure 6 shows the relationship between PA and ECW/BCM before and after applying FIR to the 27 subjects. Figure 6a shows the relationship between PA and ECW/BCM before and after applying FIR for 30 min on a loess bio-ball mat set at 40 °C. The mean value remains unchanged at 0.43 ± 0.01 [L/kg] before and after applying FIR. Figure 6b shows the relationship between PA and ECF/BCM before and after applying FIR for 7 h on a loess bio-ball mat set at 30 °C during sleep. The mean ECW/BCM value decreased by 2.33% from 0.43 ± 0.01 [L/kg] to 0.42 ± 0.01 [L/kg]. The *p*-values for PA and ECW/BCM before and after FIR application were less than 0.001. This suggests that exposure to FIR emitted from loess bio-balls during sleep can improve cell membrane function and reduce inflammatory fluid levels, thereby alleviating inflammation and edema.

## 4. Discussion

In this study, the effects of FIR emitted from loess bio-balls on bioimpedance parameters and body fluid changes were investigated using a body fluid analyzer. First, 27 subjects in group A were exposed to FIR emitted from a loess bio-ball mat set at 40 °C for 30 min. There were minimal or no changes in the bioimpedance parameters and inflammatory fluids. Second, to investigate the bioimpedance effects of FIR exposure during sleep, 27 subjects in group B were exposed to FIR emitted from a loess bio-ball mat set at 30 °C for 7 h. Changes in PA and inflammatory fluid-related parameters (ECW, ECW/ICW, ECW/TBW, and ECW/BCM) were observed following FIR exposure during sleep. These results confirm previous reports that the frequency (or wavelength) of FIR emitted from loess bio-balls overlaps with the frequency (or wavelength) of water molecules in body cells and tissues, thereby activating water molecules through selective energy absorption and improving blood and lymph circulation [1]. Exposure to FIR emitted from loess bio-balls during sleep improves lymphatic circulation and releases small amounts of waste and inflammatory body fluids in the tissue and ISF. This effectively removes waste products and reduces inflammation, which helps reduce swelling and improve overall health. As bioimpedance parameters reflect the health of cells and tissues in the body, this method may help detect inflammatory edema or pressure ulcers at an early stage and prevent further progression [12].

This study has limitations in that it was an exploratory preclinical study conducted on participants (*n* = 54) who were diagnosed with inflammation or edema at a medical institution or complained of these symptoms in the early stages of the disease. If a similar study could be replicated in a controlled clinical setting involving patients with more progressive inflammation or edema, more useful research results on inflammation and edema could be obtained.

## 5. Conclusions

The FIR emitted from loess bio-balls has a vibration frequency that overlaps with that of water molecules in the body. Therefore, the FIR emitted from loess bio-balls can activate water molecules in cells and tissues, thereby reducing inflammatory fluid and edema through lymphatic circulation. Changes in bioimpedance parameters related to inflammatory fluids in groups A and B were measured using a body water analyzer at different intensities and durations before and after FIR application. For healthy subjects, the average values of the impedance parameters measured using the body water analyzer used in this experiment are as follows: PA (6–8°), ICW (23.5–28.7 L), ECW (14.4–17.6 L), ECW/ICW (0.613), EXW/TBW (0.36–0.39), and ECW/BCM (0.428–0.429). In subjects exposed to FIR emitted from a loess bio-ball mat set at 40 °C for 30 min, there was minimal or no change in bioimpedance parameters related to PA and inflammatory fluid (ECW). However, in the experimental group exposed to FIR emitted from a loess bio-ball mat at 30 °C for 7 h during sleep, bioimpedance values such as PA (+3.80%), ECW (−3.38%), ECW/ICW (−4.84%), ECW/TBW (−2.63%), and ECW/BCM (−2.33%) related to inflammatory fluid were significantly changed.

Lymphatic circulation is a very slow response (average 0.9 mm/s) [34] because of the force exerted on the lymphatic vessel walls by surrounding tissues and the gentle contraction of the lymphatic vessels that collect lymph [4]. Therefore, the bioimpedance effects associated with early-stage inflammation and edema were further enhanced by prolonged exposure to FIR. These results suggest that lymphatic circulation may be improved by exposure to FIR during sleep, which may improve cellular function, inflammatory fluid, swelling, and edema in the body. Further studies using far-infrared rays emitted from loess bio-balls are expected to help reduce inflammation, swelling and edema, and prevent bedsores.

## Figures and Tables

**Figure 1 biomedicines-12-02392-f001:**
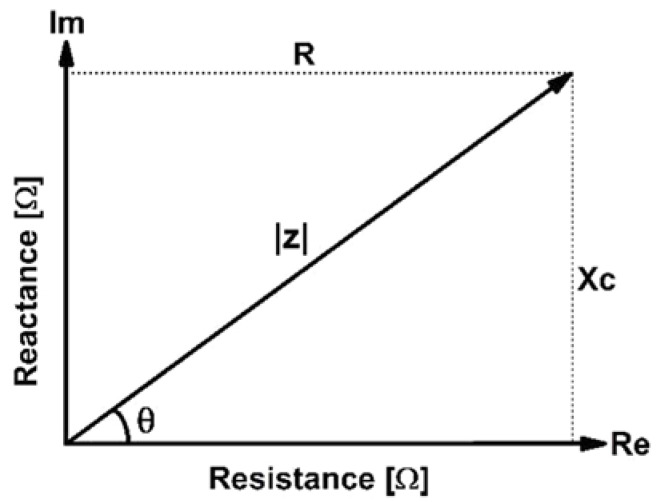
Diagram illustrating the concept of a complex impedance. *Z* is impedance, |Z| is the magnitude of impedance, *R* is the resistance, *Xc* is the reactance, and θ is PA.

**Figure 2 biomedicines-12-02392-f002:**
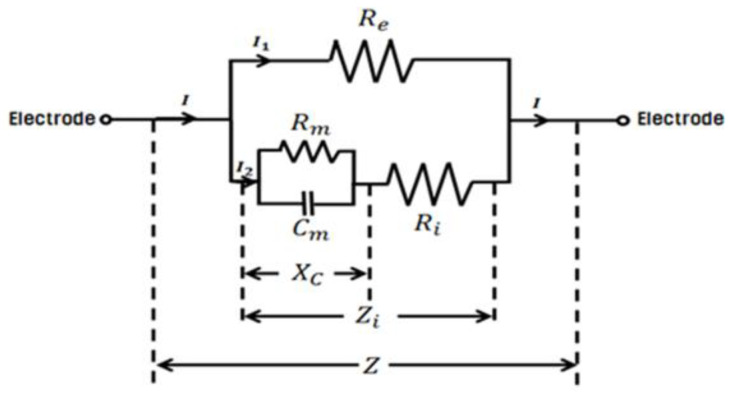
Equivalent circuit consisting of *R_e_* (ECW), cell membrane (*C_m_*), and *R_i_* (ICW). Here, *X_C_* is the reactance of the cell membrane, *Z_i_* is the impedance of *X_C_* and R_i_, and *Z* is the impedance of *Z_i_* and *R_e_*.

**Figure 3 biomedicines-12-02392-f003:**
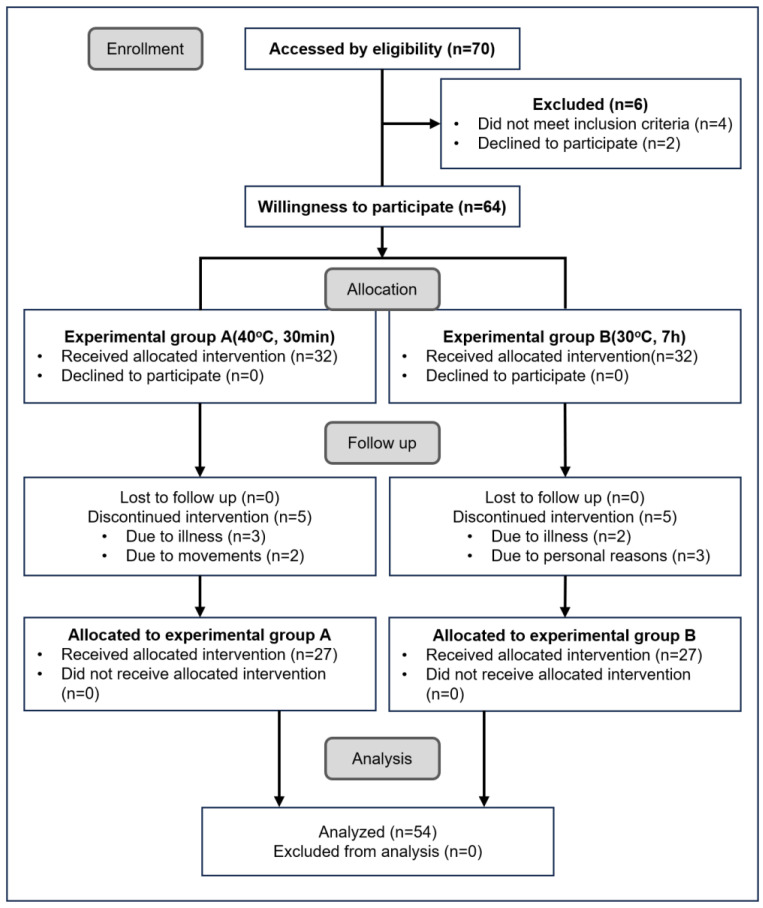
CONSORT diagram of enrollment, participation, and experimental data analysis.

**Figure 4 biomedicines-12-02392-f004:**
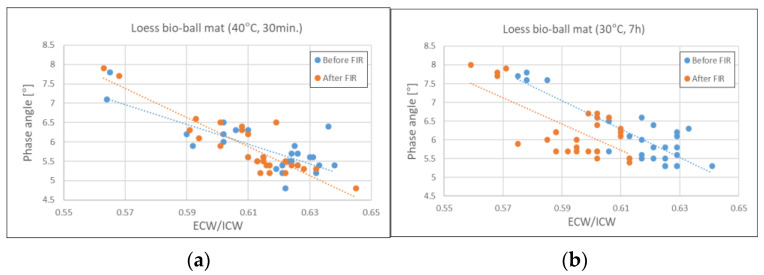
Relationship between PA and ECW/ICW before and after FIR from loess bio-ball mat: (**a**) before and after 30 min of exposure to FIR on a loess bio-ball mat set at 40 °C; (**b**) before and after 7 h of exposure to FIR on a loess bio-ball mat set at 30 °C. Circles (blue, orange) represent measured values for 27 participants, and dotted lines (blue, orange) represent estimated lines.

**Figure 5 biomedicines-12-02392-f005:**
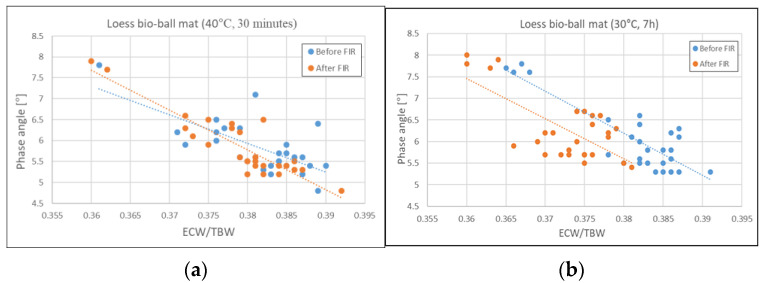
Relationship between PA and ECW/TBW before and after FIR from loess bio-ball mat. (**a**) before and after 30 min of exposure to FIR on a loess bio-ball mat set at 40 °C; (**b**) before and after 7 h of exposure to FIR on a loess bio-ball mat set at 30 °C during sleep. Circles (blue, orange) represent measured values for 27 participants, and dotted lines (blue, orange) represent estimated lines.

**Figure 6 biomedicines-12-02392-f006:**
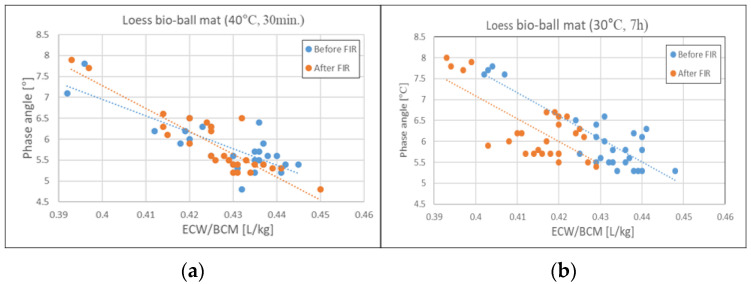
Relationship between PA and ECW/BCM before and after FIR from loess bio-ball mat: (**a**) before and after 30 min of exposure to FIR on a loess bio-ball mat set at 40 °C; (**b**) before and after 7 h of exposure to FIR on a loess bio-ball mat set at 30 °C. Circles (blue, orange) represent measured values for 27 participants, and dotted lines (blue, orange) represent estimated lines.

**Table 1 biomedicines-12-02392-t001:** Demographic characteristics and physical conditions of the study subjects (*n* = 54) in group A and group B.

Variables		Group A (*n* = 27)	Group B (*n* = 27)
Gender	Female	10 (37.04%)	15 (55.56%)
	Male	17 (62.96%)	12 (44.44%)
Age [years]		63.96 ± 5.61	60.11 ± 5.76
Height [cm]		169.15 ± 7.33	164.52 ± 9.43
Mass [kg]		63.72 ± 6.91	64.85 ± 8.96
BMI [kg/m^2^]		22.27 ± 2.44	23.89 ± 2.47
Inflammation or swelling		Stage 0, Stage 1	Stage 0, Stage 1

**Table 2 biomedicines-12-02392-t002:** Change in PA, body fluids (ICW, ECW, TBW), and body fluid-related values before and after applying FIR.

ITEMS	PA	ICW	ECW	ECW/ICW	TBW	ECW/TBW	BCM	ECW/BCM
Before FIR 40 °C, 30 min.	5.83 ± 0.63	20.74 ± 4.34	12.69 ± 2.34	0.61 ± 0.02	33.32 ± 6.53	0.38 ± 0.01	29.75 ± 6.25	0.43 ± 0.01
After FIR 40 °C, 30 min.	5.86 ± 0.74	20.87 ± 4.40	12.70 ± 2.40	0.61 ± 0.02	33.56 ± 6.79	0.38 ± 0.01	29.90 ± 6.30	0.43 ± 0.01
Delta FIR 40 °C, 30 min.	0.03 (+0.51%)	0.13 (+0.63%)	0.01 (+0.08%)	0.00 (0.00%)	0.24 (+0.72%)	0.00 (0.00%)	0.15 (+0.50%)	0.00 (0.00%)
Before FIR 30 °C, 7 h.	6.06 ± 0.79	23.19 ± 5.56	14.22 ± 3.07	0.62 ± 0.02	37.41 ± 8.62	0.38 ± 0.01	33.16 ± 7.85	0.43 ± 0.01
After FIR 30 °C, 7 h.	6.29 ± 0.76	23.19 ± 5.43	13.74 ± 3.02	0.59 ± 0.02	36.93 ± 8.43	0.37 ± 0.01	33.12 ± 7.85	0.42 ± 0.01
Delta FIR 30 °C, 7 h.	0.23 (+3.80%)	0.00 (().00%)	−0.48 (−3.38%)	−0.03 (−4.84%)	−0.48 (−1.28%)	−0.01 (−2.63%)	−0.04 (−0.12%)	−0.01 (−2.33%)

All *p*-values for the bioimpedance parameters measured before and after FIR application were less than 0.001.

## Data Availability

Data contain sensitive personal and physical information on the study participants and are available from the corresponding author upon reasonable request.

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
