# Peer review of "Effects of Far-Infrared Rays Emitted from Loess Bio-Balls on Lymphatic Circulation and Reduction of Inflammatory Fluids"

_biomedicines, 2024, doi:10.3390/biomedicines12102392_

Round 1
Reviewer 1 Report
Comments and Suggestions for Authors
Kim et al. investigated the effects of far-infrared rays emitted from loess bio-balls on lymphatic circulation and reduction of inflammatory fluids. The manuscript is well organized and the results sound good. In my view, the following minor issues should be resolved before possible acceptance by Biomedicines.
1. There seem to be too many keywords (less than 5).
2. The introduction section needs to be carefully rewritten.
3. The English throughout the manuscript should be carefully polished.
Comments on the Quality of English LanguageThe English throughout the manuscript should be carefully polished.
Author Response
Manuscripts must be written in good English and properly edited.
The paper was finally revised and edited by a native English-speaking co-author (min Seok Kim, MD).
Introduction: Include sufficient background and all appropriate references in the introduction.
56-94: In accordance with the reviewer's comments, 2-2 and 2-3 were included in the introduction
101-116: Related literature was cited to describe the relationship between bioimpedance parameters and inflammatory fluid-related indicators applied in medical settings.
Response to Comment
- Too many keywords (less than 5)
31: Keywords have been reduced to 5.
- Introduction section needs to be carefully rewritten.
34-116: The introduction has been carefully rewritten to include relevant information for readers' ease of understanding.
- English should be carefully polished.
In the paper, English sentences were revised by native-speaking co-authors.

Reviewer 2 Report
Comments and Suggestions for Authors
Far infrared rays (FIR) have shown considerable promise for the investigation of lymphatic circulation and treatment of inflammation. The manuscript introduces a new study to understand how body fluids change based on the intensity and exposure time of FIR emitted from loess bio-balls. The results not only highlight the potential of FIR therapy in reducing inflammatory fluids but also its applicability in improving lymphatic circulation with extended exposure during sleep. Overall, this study presents an approach that demonstrates the gradual reduction of inflammatory fluids through long-term FIR exposure. Considering the following suggestions, I recommend the manuscript after minor revisions.
1. The authors should also compare different materials that emit FIR and discuss the advantages of using loess bio-balls for FIR therapy.
2. The authors should elaborate on why the BIA method can comprehensively reflect lymphatic circulation and inflammation in patients.
3. Regarding the experimental design, why does Group A have a higher temperature than Group B? Why were the temperatures set at 40°C and 30°C?
Author Response
Corrections in response to reviewer 2's comments
Manuscripts must be written in good English and properly edited.
The paper was finally revised and edited by a native English-speaking co-author (Min Seok Kim, MD).
Introduction: Include sufficient background and all appropriate references in the introduction.
56-94: In accordance with the reviewer's comments, 2-2 and 2-3 were included in the introduction
101-116: Related literature was cited to describe the relationship between bioimpedance parameters and inflammatory fluid-related indicators applied in medical settings.
Conclusions should be supported and improved by experimental results.
384-390: The conclusion was supported by experimental results.
Response to Comment
- Discuss the advantages of using loess bio-balls for FIR therapy, using other substances that emit FIR.
Bioimpedance measurements were performed on 22 participants using conventional electric mats and high-temperature heat-treated loess balls and stone beds, but there was little change related to inflammatory fluids.
Impedance measurements using carbon graphene will be used to investigate changes associated with inflammatory fluids, which will be addressed in a subsequent study.
- The authors should elaborate on why the BIA method can comprehensively reflect lymphatic circulation and inflammation in patients.
BIA provides many parameters reflecting inflammatory fluids that can be applied to a variety of patients in the medical setting. In the paper, these contents were described as follows: 96-116, 205-242, 244-252, 262-267, 282-310, 312-317, 333-338.
- In the experimental design, why were the temperatures set to 40 and 30 °C for the loess bio-ball mats?
The loess bio-ball emits large amounts of FIR starting at 40 °C, so the FIR treatment effect is measured at 40 °C under normal conditions. However, the temperature was lowered to 30 °C during sleep so that the subjects could lie down comfortably and fall asleep.

Reviewer 3 Report
Comments and Suggestions for Authors
The study examined how far infrared rays (FIR) from loess bio-balls affect body fluids and inflammation. Two groups were exposed to different FIR intensities and durations. Group A (30 min) showed minimal changes, while Group B (7 hours) showed significant improvements in fluid-related parameters, suggesting that FIR can reduce inflammation by improving lymphatic circulation.
Major comments:
1. Could authors provide more explanation and references about how the bioimpedance changing relates to the inflammatory fluid (not health body fluid) level in vivo in the introduction section?
2. The section 2.2 and 2.3 are background information, how about moving these to introduction section?
3. What are the criteria for exclusion or discontinued intervention in Figure 3?
4. Line 203-205: Do these differences have any physiological changes to patients? Do authors have any data or reference to show such differences have any significant biological effect? How does these parameters change after the conventional therapy?
5. I fully understand adding control groups is extremely difficult in human experiments, but could authors provide any reference about those parameters (PA, ICW, ECW, ECW/ICW, TBW, ECF/TBW, BCM, ECW/BCM) of health people?
Minor comments:
1. I think the results in Figure 4 and 5 have been shown in the table 2. I suggest keeping only one way to demonstrate these results.
2. Please add p values in the figures or table 2.
3. Line 167-168: The treatment was conduct to whole body or just swelling sites?
Author Response
Corrections in response to reviewer 3's comments
Manuscripts must be written in good English and properly edited.
The paper was finally revised and edited by a native English-speaking co-author (Min Seok Kim, MD).
Major Comments
- Include sufficient background and all appropriate references in the introduction.
101-116: Related literature was cited to describe the relationship between bioimpedance parameters and inflammatory fluid-related indicators applied in medical settings.
- Move Sections 2.2 and 2.3 to Introduction.
56-94: In accordance with the reviewer 3's comments, 2-2 and 2-3 were included in the introduction.
- Describe the criteria for exclusion or discontinued intervention in Figure 3.
164-168: The exclusion or discontinuation criteria for the intervention were described in the text for Figure 3.
- In lines 203-205, do these differences bring about physiological changes in the patient?
Various bioimpedance parameters are applied to various diseases in the medical field, and the changes in bioimpedance parameters due to treatment effects indicate physiological effects. These parameters are reflected even after treatment, and many studies (data and references) are reported on this.
References: 20, 21, 22, 26, 27, 28, 31, 32, 33, 36, 41, 42
- It is very difficult to add a control group in human trials, but could the authors provide references for parameters in healthy people?
For control group, bioimpedance measurements were performed on 22 participants using conventional electric mats and high-temperature heat-treated loess balls and stone beds, but there was little change related to inflammatory fluids.
The impedance parameters of healthy subjects, as measured by the body water meter used in this experiment, are as follows. ICW(23.5-28.7L), ECW(14.4-17.6L), ECW/ICW(0.613), ECW/TBW(0.36-0.39), ECW/BCM(0.428-0.429).
Minor comments
- The results of Figures 4 and 5 are shown in Table 2. Only one method that shows these results should be maintained.
241-242: Figures 4 and 5 were removed and Table 2 was modified to make it easier for readers to understand.
- Insert the p-values into the figures and Table 2.
242, 299-300, 324-325, 345-346: P -values are inserted in Table 2 and text for Figures 4, 5, and 6
- Line 167-168, Does the treatment be conducted to whole body or just swelling sites?
179-183: FIR emitted from loess bio-ball was exposed on whole body with clamp-type electrodes attached to both ankles and middle fingers.

Round 2
Reviewer 3 Report
Comments and Suggestions for Authors
Most of comments were addressed, but I suggest adding the impedance parameters of healthy subjects into the result section (Major comment 5).
Author Response
Most of comments were addressed, but I suggest adding the impedance parameters of healthy subjects into the result section (Major comment 5).
Based on Reviewer 3's comments and suggestions, the following was added to the results section.
384-387: For healthy subjects, the average values of the impedance parameters measured using the body water analyzer used in this experiment are as follows: PA (6-8°), ICW (23.5-28.7 L), ECW (14.4-17.6 L), ECW/ ICW (0.613), EXW/TBW (0.36-0.39), and ECW/BCM (0.428-0.429).
